# Saturated Fatty Acid Blood Levels and Cardiometabolic Phenotype in Patients with HFpEF: A Secondary Analysis of the Aldo-DHF Trial

**DOI:** 10.3390/biomedicines10092296

**Published:** 2022-09-15

**Authors:** Katharina Lechner, Clemens von Schacky, Johannes Scherr, Elke Lorenz, Matthias Bock, Benjamin Lechner, Bernhard Haller, Alexander Krannich, Martin Halle, Rolf Wachter, André Duvinage, Frank Edelmann

**Affiliations:** 1Rehabilitation and Sports Medicine, Department of Prevention, School of Medicine, Technical University of Munich, 80992 Munich, Germany; 2DZHK (German Centre for Cardiovascular Research), Partner Site Munich, Munich Heart Alliance, 80336 Munich, Germany; 3Kardiologie, Deutsches Herzzentrum München, 80636 Munich, Germany; 4Omegametrix, Martinsried, 82152 Munich, Germany; 5University Center for Prevention and Sports Medicine, Balgrist University Hospital, University of Zurich, 8008 Zurich, Switzerland; 6Department of Internal Medicine IV, Ludwig-Maximilians University, 80336 Munich, Germany; 7Institute of AI and Informatics in Medicine, Klinikum rechts der Isar, Technische Universität München, 81675 Munich, Germany; 8Charité, Universitätsmedizin Berlin, 10117 Berlin, Germany; 9Clinic and Policlinic for Cardiology, University Hospital Leipzig, 04103 Leipzig, Germany; 10Department of Cardiology and Pneumology, University Medical Center Göttingen, Georg-August University, 37077 Göttingen, Germany; 11DZHK (German Centre for Cardiovascular Research), Partner Site Göttingen, 37075 Göttingen, Germany; 12Department of Cardiology, Charité, Universitätsmedizin Berlin, 10117 Berlin, Germany; 13DZHK (German Centre for Cardiovascular Research), Partner Site Berlin, 10785 Berlin, Germany

**Keywords:** saturated fatty acids, C14:0, C16:0, VLSFAs, heart failure, HFpEF, diastolic dysfunction, metabolic phenotype, atherogenic dyslipidemia, aerobic capacity

## Abstract

Background: Circulating long-chain (LCSFAs) and very long-chain saturated fatty acids (VLSFAs) have been differentially linked to risk of incident heart failure (HF). In patients with heart failure with preserved ejection fraction (HFpEF), associations of blood SFA levels with patient characteristics are unknown. Methods: From the Aldo-DHF-RCT, whole blood SFAs were analyzed at baseline in *n* = 404 using the HS-Omega-3-Index^®^ methodology. Patient characteristics were 67 ± 8 years, 53% female, NYHA II/III (87%/13%), ejection fraction ≥50%, E/e’ 7.1 ± 1.5; and median NT-proBNP 158 ng/L (IQR 82–298). Spearman´s correlation coefficients and linear regression analyses, using sex and age as covariates, were used to describe associations of blood SFAs with metabolic phenotype, functional capacity, cardiac function, and neurohumoral activation at baseline and after 12-month follow-up (12 mFU). Results: In line with prior data supporting a potential role of de novo lipogenesis-related LCSFAs in the development of HF, we showed that baseline blood levels of C14:0 and C16:0 were associated with cardiovascular risk factors and/or lower exercise capacity in patients with HFpEF at baseline/12 mFU. Contrarily, the three major circulating VLSFAs, lignoceric acid (C24:0), behenic acid (C22:0), and arachidic acid (C20:0), as well as the LCSFA C18:0, were broadly associated with a lower risk phenotype, particularly a lower risk lipid profile. No associations were found between cardiac function and blood SFAs. Conclusions: Blood SFAs were differentially linked to biomarkers and anthropometric markers indicative of a higher-/lower-risk cardiometabolic phenotype in HFpEF patients. Blood SFA warrant further investigation as prognostic markers in HFpEF. One Sentence Summary: In patients with HFpEF, individual circulating blood SFAs were differentially associated with cardiometabolic phenotype and aerobic capacity.

## 1. Introduction

Heart failure (HF) with preserved ejection fraction (HFpEF) is a heterogeneous clinical condition with a number of underlying etiologies [1]. Its prevalence continues to rise collinear to the aging population and/or risk factors such as (central) obesity, type 2 diabetes mellitus (T2D), and hypertension [2]. Obesity-related HFpEF is an important phenotype present in the subgroup of the population with metabolic disorders such as T2D [3], and prognostic outcome in these individuals is largely determined by comprehensive treatment of comorbidities such as low aerobic capacity and/or cardiovascular risk factors [3,4].

Saturated fatty acids (SFAs) are a heterogeneous group of fatty acids that occur in a variety of foods including whole-fat dairy, meat, cocoa, and industrially processed foods [5,6]. They are, furthermore, endogenously synthesized by de novo lipogenesis (DNL) in the liver, a metabolic pathway that converts dietary starch, sugar, protein, and alcohol into fatty acids (FAs), in the presence of nutrient overabundance and/or (hepatic) insulin resistance [7,8,9].

Individual SFAs, as assessed in different lipid compartments in the body such as plasma or erythrocytes, have been differentially linked to incident HF [10,11,12], clinical traits associated with the obesity-related HFpEF phenotype such as T2D and/or atrial fibrillation [13,14], and cardiovascular endpoints [15] in previous analyses. In this regard, higher circulating levels of some SFAs such as palmitic acid (PA, C16:0) have been linked to increased risk of developing T2D [16], to higher risk of incident HF [17], and to higher risk of mortality in patients referred for coronary angiography [15]. Conversely, higher levels of other SFAs, such as the three very long-chain saturated fatty acids (VLSFAs) arachidic acid (AA, C20:0), behenic acid (BA, C22:0), and lignoceric acid (LA, C24:0), have been broadly linked to lower risk of these outcomes [10,13] (all in comparison to lower levels), which overall supports the notion that SFAs are a biologically heterogeneous group of fatty acids.

SFAs biomarkers such as red blood cell (RBC) SFAs—and by extension whole blood SFAs—reliably reflect cardiac and other tissue SFA levels over the preceding three months [18,19] (i.e., the balance of intake, endogenous production, distribution volume, and catabolism) independent of subjective memory-based assessment methods such as food-frequency questionnaires [20]. In the latter regard, it is important to acknowledge that overabundance of SFAs such as C16:0, which have been linked to adverse health effects, seems to be more related to endogenous overproduction due to energy excess and/or metabolic conditions such as non-alcoholic fatty liver disease (NAFLD), T2D, and MetS than to dietary uptake [8,16,17].

In patients with HFpEF, the association of blood SFA levels and phenotypic traits is not known. To fill this gap, we report individual SFA whole blood levels in a large cohort comprised of 404 HFpEF patients and associations with cardiometabolic phenotype, functional capacity, echocardiographic markers indicative of left ventricular diastolic function (LVDF), and neurohumoral activation in the framework of the Aldosterone in Diastolic Heart Failure (Aldo-DHF) trial.

## 2. Methods

### 2.1. Study Design

This is a post hoc analysis of the Aldo-DHF trial (ISRCTN 94726526). We analyzed associations of whole blood SFA levels at baseline with patient characteristics at baseline and after 12 months (12 mFU). From a total of 422 patients enrolled in the Aldo-DHF trial, 18 whole blood aliquots were not available due to loss during storage/transfer or missing blood sampling at baseline.

### 2.2. Aldo-DHF Trial

The Aldo-DHF trial was a multicenter, prospective, randomized, double-blind, and placebo-controlled trial that evaluated the effect of a 12-month aldosterone receptor blockade on diastolic function (E/e’) and maximal exercise capacity (VO2peak) in patients with HFpEF. Participants were eligible if they were men and women aged 50 years or older with current HF symptoms consistent with New York Heart Association (NYHA) class II or III, had left ventricular ejection fraction (LVEF) of 50% or greater, had echocardiographic evidence of diastolic dysfunction (grade I) or atrial fibrillation at presentation, and had maximum exercise capacity (VO2peak) of 25 mL/kg/min or less [21]. Exclusion criteria have been published before [21]. In total, 422 patients (mean age 67 (SD, 8) years; 52% female) with evidence of diastolic dysfunction were included. Data acquisition took place between March 2007 and April 2012 at 10 sites in Germany and Austria [21].

#### 2.2.1. Laboratory Measurements

In the Aldo-DHF Trial, venous blood samples were drawn after 20 min of rest in supine position under standardized conditions. Samples were immediately cooled and processed for storage at −80 °C (−112 °F). N-terminal pro–brain-type natriuretic peptide (NT-proBNP) was analyzed with the Elecsys NT-proBNP immunoassay (Roche Diagnostics) [21].

The process of immediate freezing and storage at –80 °C of the blood samples from the Aldo-DHF trial resulted in stable fatty acid levels [22]. For gas chromatographic analysis of fatty acid composition, 2.0 mL aliquots of frozen (−80 °C) EDTA-blood were shipped to a reference laboratory for fatty acid analyses (Omegametrix, Martinsried, Germany). At Omegametrix, whole blood fatty acid composition was analyzed according to the HS-Omega-3 Index^®^ methodology, as previously described [23]. Fatty acid methyl esters were generated by acid transesterification and were analyzed by gas chromatography using a GC2010 Gas Chromatograph (Shimadzu, Duisburg, Germany) equipped with a SP2560, 100 m column (Supelco, Bellefonte, PA, USA) using hydrogen as carrier gas. Fatty acids were identified by comparison with a standard mixture of fatty acids characteristic of erythrocytes. Individual fatty acid results are given as relative amounts of myristic acid (C14:0), palmitic acid (C16:0), stearic acid (C18:0), arachidic acid (C20:0), behenic acid (C22:0) and lignoceric acid (C24:0) expressed as a percentage of a total of 26 identified fatty acids in whole blood. Analyses were quality-controlled according to DIN ISO 15189.

#### 2.2.2. Echocardiography and Other Variables

In the Aldo-DHF Trial, clinical data were obtained and diagnostic procedures were completed according to predefined standard operating procedures based on international guidelines [21]. Diastolic function on echocardiography was assessed in accordance with American Society of Echocardiography guidelines [24].

### 2.3. Ethics

The Aldo-DHF Trial complied with the Declaration of Helsinki and principles of good clinical practice. The study protocol was approved by the responsible ethics committees (approval code 6/12/06; date 25 February 2007). All participants gave written informed consent prior to any study-related procedures.

### 2.4. Statistical Analysis

Continuous variables are reported as mean +/− standard deviation (SD) or median (interquartile range (IQR)), according to their scale and distribution. Categorical variables are presented as absolute and relative frequencies. Spearman’s correlation coefficients and multiple linear regression analyses, using sex and age as covariates, were used to describe the association of individual SFAs (C14:0 (myristic acid), C16:0 (palmitic acid), C18:0 (stearic acid), C20:0 (arachidic acid), C22:0 (behenic acid), C24:0 (lignoceric acid)) with cardiometabolic risk markers, echocardiographic markers of left ventricular diastolic function, and neurohumoral activation at baseline and at 12 mFU. To account for the randomization group, all analyses were repeated as sensitivity analysis with group as covariate. Further, a principal component analysis (PCA) was conducted to consolidate variables due to the limited sample size. A significance level of α = 5% was used for all tests. All tests were hypothesis-generating without confirmatory interpretation. Therefore, no correction was applied to counteract the problem of multiple comparisons. All statistical analyses were performed using IBM SPSS Statistics for Windows, version 25 (IBM Corp., Armonk, NY, USA) as well as R and RStudio, version R-4.2.125 (R Foundation for Statistical Computing, Vienna, Austria).

## 3. Results

### 3.1. Study Population

Baseline characteristics are shown in Table 1.

Associations between individual saturated fatty acids and patient characteristics are shown in Table 2 and Table 3. Results of the linear regression models are reported in the main text.

Figure 1 shows correlation plots at baseline and 12 mFU.

### 3.2. Long Chain Fatty Acids

#### 3.2.1. Myristic Acid (MA, C14:0)

The long-chain SFA myristic acid (MA, C14:0) was positively associated with LDL-C (r = 0.145, *p* = 0.005), non-HDL-C (r = 0.234, *p* = <0.001) and markers for diabetic dyslipidemia such as triglycerides-to-HDL-C ratio (r = 0.393, *p* = <0.001; β = 5.45, *p* = <0.001) and triglycerides (r = 0.428, *p* = <0.001; β = 205.14, *p* = <0.001). These associations persisted after 12 mFU. Additionally, after 12 mFU, HbA1c (r = 0.146, *p* = 0.007) and anthropometric markers for truncal adiposity such as waist-to-height ratio (r = 0.117, *p* = 0.031) showed a direct association with C14:0. Higher C14:0 was, furthermore, predictive of lower submaximal aerobic capacity (β = −65.51, *p* = <0.001) at 12 mFU.

#### 3.2.2. Palmitic Acid (PA, C16:0)

Similar to C14:0, the long-chain SFA palmitic acid (PA, C16:0) was positively associated with dyslipidemia, as depicted in Figure 2, and surrogate markers for non-alcoholic fatty liver disease, as depicted in Table 2. Additionally, it was directly associated with HbA1c (β = 0.09, *p* = <0.001) and inversely associated with submaximal aerobic capacity (r = −0.14, *p* = 0.007; β = −8.17, *p* < 0.001) at baseline. Higher C16:0 at baseline was associated with dyslipidemia, surrogate markers for non-alcoholic fatty liver disease, central adiposity and lower submaximal aerobic capacity (β = −12.66, *p* = <0.001), and lower maximal aerobic capacity (β = −0.27, *p* = 0.019) at 12 mFU.

#### 3.2.3. Stearic Acid (SA, C18:0)

The long-chain SFA stearic acid (SA, C18:0) was inversely associated with triglycerides-to-HDL-C ratio (β = −0.38, *p* = <0.001), triglycerides (β = −16.85, *p* = <0.001), non-HDL-C (β = −6.45, *p* = <0.001), and LDL-C (β = −5.71, *p* = <0.001) at baseline. Higher blood levels of C18:0 at baseline were associated with a more favorable lipid phenotype at 12 mFU (*p* = <0.001 for markers of diabetic dyslipidemia using linear regression models).

### 3.3. Very Long Chain Fatty Acids

Higher blood levels of the three long-chain SFAs, arachidic acid (AA, C20:0), behenic acid (C22:0), and lignoceric acid (C24:0), were all associated with a lower triglycerides-to-HDL-C ratio, lower triglycerides, lower non-HDL-C, and lower LDL-C (the latter only in the linear regression models) at baseline and after 12 mFU.

#### 3.3.1. Arachidic Acid (AA, C20:0)

The very long-chain SFA arachidic acid (AA, C20:0) was inversely associated with BMI and central adiposity, alanine aminotransaminase, and γ-glutamyltransferase (β = −261.57, *p* = <0.01) at baseline and additionally associated with a higher VO2peak (β = 17.69, *p* = 0.009) at 12 mFU.

#### 3.3.2. Behenic Acid (C22:0)

In addition to its association with a more favorable lipid profile (lower non-HDL-C, triglycerides, and triglycerides-to-HDL-C ratio) and higher VO2peak, as depicted in Table 2, higher levels of behenic acid (C22:0) were inversely associated with γ-glutamyltransferase (β = −37.02, *p* = 0.022) at baseline in linear regression models.

#### 3.3.3. Lignoceric Acid (C24:0)

Lignoceric acid (C24:0) was inversely associated with LDL-C, non-HDL-C, triglycerides, triglycerides-to-HDL-C ratio (Figure 3), and γ-glutamyltransferase (β = −47.67, *p* = <0.001)]. Furthermore, higher blood levels of C24:0 were associated with higher submaximal and maximal aerobic capacity.

No pattern of significant associations was found between cardiac function and concentrations of blood LCSFAs and VLSFAs.

### 3.4. Sensitivity Analyses

Sensitivity analyses with group as covariate showed significant effects of group allocation (spironolactone^+/−^) for the 12 mFU outcomes of systolic/diastolic blood pressure (*p* = <0.001), heart rate (only C16:0), E/e′, and HbA1c but not for the markers reported above, such as blood lipids, liver enzymes, BMI/central adiposity, or aerobic capacity.

### 3.5. Sex-Specific Analyses

All models were adjusted using sex as a covariate and overall, since sex had a significant influence in several models while not turning the results in a specific direction regarding the fatty acid. In addition, sex-specific analyses within the gender subgroups were completed for all outcomes:HbA1c: In women, but neither in men nor in the entire cohort, the long-chain SFA C14:0 (MA, C14:0) was consistently directly associated with HbA1c at baseline (β = 0.48, *p* = 0.004) and at 12 mFU (β = 0.41, *p* = 0.009).Lipid phenotype: In women (and in the entire cohort, as described above), but not in men, the long-chain SFA stearic acid (SA, C18:0) was inversely associated with non-HDL-C (β = −9.11, *p* = 0.001) and LDL-C (β = −8.52, *p* = <0.001) at baseline and at 12 mFU (non-HDL-C (β = −9.33, *p* = 0.001), LDL-C (β = −7.66, *p* = 0.001)).Submaximal aerobic capacity: The very long-chain SFA lignoceric acid (C24:0) was predictive of higher distance covered in 6MWT at 12 mFU in the entire cohort (β = 64.86, *p* = 0.019) and in men (β = 91.55, *p* = 0.014) but not in women.

### 3.6. Principal Component Analysis

A PCA was conducted, but the sum of variance of the first two principal components (PCs) was relatively low at 49.2%. Therefore, these PCs were not used for any further analysis. However, the first and second PCs are shown as a biplot to visualize the similarities and differences between the used predictors in Figure 4. The PCA shows that there are two variable groups, C14:0 and C16:0, as well as C18:0, C20:0, C22:0, and C24:0, which have a similar information content within the group. This means that C14:0 and C16:0 contain similar information in the data set. Further, C18:0, C20:0, C22:0, and C24:0 are similar regarding their information content. This is the same for age and sex.

## 4. Discussion

### 4.1. Main Findings

We analyzed the associations of individual whole blood SFA proportions with cardiometabolic risk factors, exercise capacity, echocardiographic markers of left ventricular diastolic function, and neurohumoral activation in the well-phenotyped Aldo-DHF cohort, comprising 404 patients with HFpEF.

Our main finding is that individual blood SFAs, a standardized analytical biomarker for SFA levels with low analytical variability, are differentially related with cardiometabolic phenotype and aerobic capacity but not with echocardiographic markers of left ventricular diastolic function, in patients with HFpEF. We observed that individual blood SFAs had opposing associations with cardiometabolic phenotype in HFpEF patients: conglomeration 1, comprised of C14:0 and C16:0, which are both markers of endogenous fatty acids synthesis in the context of nutrient overabundance, was associated with a higher-risk cardiometabolic phenotype. Conversely, conglomeration 2, comprised of the three very long-chain saturated fatty acids (C20:0, C22:0, and C24:0) and the LCSFA C18:0, was associated with a less pronounced risk profile in HFpEF patients. These findings question the use of generalizing umbrella terms such as ‘‘saturated fatty acids’’.

### 4.2. Individual SFAs and Patient Characteristics

#### 4.2.1. SFA Conglomeration 1 (C14:0 and C16:0)

Blood long-chain saturated fatty acids (LCSFAs) reliably reflect cardiac and other tissue SFA levels over the preceding three months (i.e., the balance of intake, endogenous production, distribution volume, and catabolism). Mean whole blood percentages of C14:0 (myristic acid) and C16:0 (palmitic acid), expressed as a percentage of the total of 26 identified FAs in whole blood in the Aldo-DHF cohort, were 0.69% and 24.89% respectively. There are no comparable data available on whole blood SFA concentrations in HFpEF patients, according to our literature search. In a cohort of patients with HF, Berliner et al. reported comparable whole blood concentrations of C16:0 (22.56 ± 1.76 % of total FA in whole blood) to the levels observed in our HFpEF population [25]. C14:0 was not reported in this analysis [25]. We observed a positive association of the LCSFAs C14:0 and C16:0 with established risk factors for cardiovascular disease and/or HF such as triglycerides and non-HDL-C [26], with HbA1c [3], with triglycerides-to-HDL-C ratio, a metabolic marker for plaque phenotype (i.e., thin-cap fibroatheromas) [27,28,29], and with anthropometric markers indicative of high cardiometabolic risk such as waist-to-height ratio [30]. The liver enzymes alanine aminotransaminase and γ-glutamyltransferase, which are surrogate markers of NAFLD and MetS, were directly associated with C16:0 [31]. These results are in line with biologic plausibility. C16:0 is the most abundant SFA in the human body, accounting for 20%–30% of total FAs [8]. It serves physiological functions related to membrane physical properties, protein palmitoylation, palmitoylethanolamide (PEA) biosynthesis, and efficient pulmonary surfactant activity [8]. C16:0 comes from the diet and is synthesized endogenously via DNL [8]. Its homeostasis is physiologically tightly controlled. However, certain conditions such as the presence of a positive energy balance, excessive intake of carbohydrates (in particular, fructose [32]), a sedentary lifestyle, and medical conditions related to these risk factors such as NAFLD, MetS, and T2DM may strongly induce DNL, resulting in increased tissue content of C16:0 [7,8]. In line with these physiological considerations, sugar-sweetened beverage consumption (i.e., dietary fructose intake) was consistently positively associated with higher concentrations of C16:0 ceramides in the Framingham Offspring Cohort [33]. High levels of C16:0 can disrupt tissue membrane phospholipid balance, which may depend on an optimal ratio of C16:0 with unsaturated fatty acids, especially n-3 and n-6 polyunsaturated fatty acids (PUFAs) [8]. Therefore, aligning with our findings, overaccumulation of C16:0 in tissues [8] and enrichment of C16:0 in very low-density lipoprotein particles (VLDL-P) but not in plasma [34] has been linked to dyslipidemia, pancreatic β-cell dysfunction/hyperglycemia, increased ectopic fat accumulation, and re-emergence of T2D in relapsers in a vicious cycle. Furthermore, C16:0 contributes to systemic and vascular inflammation through dimerization and activation of toll-like receptor (TLR) 2/4 [35]. In our analysis, higher C16:0 was, furthermore, predictive of lower submaximal aerobic capacity, a risk indicator for adverse outcomes in HFpEF, but there are no previous data on C16:0 and functional capacity to the best of our knowledge. C14:0 is another marker of DNL. Aligning with our observation of a positive association of C14:0 and triglycerides, increasing concentrations of C14:0 have been linked to progressive increases in triglycerides and ApoCIII concentrations, a hallmark of inflammatory potential of low-density lipoproteins [36], independently of coronary artery disease (CAD) diagnosis and gender in plasma samples from 1370 subjects with or without angiographically demonstrated CAD [37].

Overall, in line with physiological considerations (i.e., C16:0 as a marker of DNL in the context of nutrient excess) [8] and previous analyses in HF [17] and in ASCVD [15], C16:0—compared to other SFAs—was most consistently associated with a higher risk cardiometabolic phenotype. Interestingly, in 424 subjects from the PREDIMED randomized dietary trial, participants in the virgin olive oil and nut group showed increased plasma concentrations of C16:0 after completing a 1-year intervention program in the context of stable weight. [38] However, the methodology used (plasma concentrations) does not reflect long term dietary intake [38] and is, therefore, not directly comparable to the methodology used in our analysis.

#### 4.2.2. SFA Conglomeration 2 (C18:0 and Very Long-Chain Saturated Fatty Acids)

Whole blood very long-chain saturated fatty acids (VLSFAs) are biomarkers of metabolism (i.e., elongation of LCSFAs) and, to a lesser extent, dietary intake. The mean whole blood percentages of C20:0 (arachidic acid), C22:0 (behenic acid), and C24:0 (lignoceric acid) expressed as a percentage of a total of the 26 identified FAs in whole blood in this cohort were 0.18%, 0.5% and 0.75%, respectively. The mean whole blood percentage of C18:0 (stearic acid) was 13.87%. No comparable data are available on whole blood SFA concentrations in HF patients, according to our literature search. Berliner et al. reported slightly lower whole blood concentrations of C18:0 (10.80 ± 2.08% of total FA in whole blood) and C24:0 (0.22 ± 0.10% of total FA in whole blood) in a cohort of HFrEF patients than the levels observed in our HFpEF population. [25] Notably, C20:0 and C22:0 were not reported by Berliner et al. [25].

Stearic acid (C18:0), the second-most-abundant SFA after C16:0, is derived from the diet (e.g., from full-fat dairy such as cheese and from meat) [5], through elongation from C16:0, and serves as a substrate for the synthesis of VLSFAs that are produced from C18:0 by elongases [13]. Contrary to C14:0 and C16:0, we observed that higher blood levels of the long-chain SFA C18:0 were broadly associated with a more favorable lipid profile (lower triglycerides-to-HDL-C ratio, triglycerides, non-HDL-C, and LDL-C) at baseline and at 12 mFU. Higher whole blood C18:0 was, furthermore, associated with lower liver enzymes (aspartate aminotransaminase and γ-glutamyltransferase) at 12 mFU in the entire cohort, suggesting an overall beneficial association of whole blood C18:0 with biomarkers of the DNL pathway.

In our cohort, the most consistent inverse association of whole blood VLSFAs was observed with lipid risk markers, in particular triglycerides. In line with this finding, Zhao et al. reported that plasma levels of the VLCSFAs C20:0, C22:0, and C24:0 were significantly and inversely associated with risk of MetS and individual components of MetS, in particular triglycerides, in 1729 Chinese adults aged 35–59 years [39]. Regarding individual VLSFAs, C20:0 was inversely associated with triglycerides-to-HDL-C ratio, triglycerides, non-HDL-C, LDL-C, anthropometric markers indicative of (central) adiposity, blood pressure, alanine aminotransaminase, and γ-glutamyltransferase at baseline and 12 mFU and, furthermore, was predictive of higher maximal aerobic capacity (VO2peak) at 12 mFU. The VLSFA C22:0 was inversely associated with triglycerides-to-HDL-C ratio, triglycerides, and non-HDL-C. The VLSFA SFA C24:0 was inversely associated with triglycerides-to-HDL-C ratio, triglycerides, non-HDL-C, and γ-glutamyltransferase at baseline and 12 mFU and predictive of a higher distance covered in 6MWT at 12 mFU.

No consistent significant associations and/or specific clusters were found between cardiac function and/or neurohumoral activation and blood SFA concentrations in this analysis, consistent with our previous analysis investigating the associations of individual Omega-3 FAs and echocardiographic markers of left ventricular diastolic function [23].

## 5. Strengths and Limitations

This analysis has limitations. First, all Aldo-DHF participants were of Caucasian origin and our findings may, therefore, not be representative for a random population sample or applicable to other ethnicities. Second, FA measurements were done only once in baseline samples. Proportions of whole blood SFAs may vary over time due to dietary changes, lifestyle changes, or diseases—in other words, they reflect a balance of influx (i.e., dietary factors or metabolic factors such as the DNL pathway) and efflux (i.e., fasting), but nothing is known about the relative determinants driving the blood SFA concentrations in this population. Third, we do not have data on prognosis, i.e., clinical endpoints. Finally, we do not have data on odd-chain SFAs (C15:0 and C17:0), which are markers for, e.g., dairy-fat intake, and which have been linked to improved mitochondrial function [6] and lower risk for T2D [40].

A major strength of our analysis is the precise clinical and metabolic characterization of the Aldo-DHF cohort comprising 404 HFpEF patients (with follow-up data after one year). Furthermore, with a proportion of 53% of the patients included in Aldo-DHF being female, this analysis adequately reflects the gender distribution in HFpEF [1]. Finally, this is the first analysis of whole blood SFA in patients with HFpEF. Whole blood FAs, as a biomarker of FA intake and endogenous production, offer a number of advantages over the assessment of FA intake via subjective methods such as food-frequency questionnaires and measurement of SFA in other lipid compartments such as plasma; regarding the latter, blood is easily accessible, whole blood levels have low biological variability, and the use of the HS-Omega-3 Index^®^ methodology provides low analytical variability [19]. Furthermore, compared to assessing the relations between FAs and surrogate markers for risk and/or clinical endpoints by subjective memory-based methods, a concept that has been questioned due to the high percentage of implausible data generated [20], measuring biomarkers is more biologically accurate.

## 6. Translational Outlook

Prognosis in HFpEF is determined by optimal risk factor control and treatment of comorbidities [4]. We found that individual blood SFAs could be broadly categorized into two main SFA conglomerations, which are differentially linked to biomarkers and anthropometric markers indicative of higher-/lower-risk cardiometabolic phenotype in HFpEF patients. In line with prior data supporting a potential role of DNL and/or DNL-related LCSFAs in the development of HF [17], we showed that baseline blood levels of C14:0 and C16:0, which are both markers for DNL [8,16], were associated with a more pronounced risk profile in the Aldo-DHF cohort at baseline and after one year. Contrarily, the three major circulating VLSFAs, arachidic acid (C20:0), behenic acid (C22:0), and lignoceric acid (C24:0), as well as the LCSFA C18:0, were broadly associated with a lower-risk HFpEF phenotype. In particular, the association of C16:0 and the higher-risk phenotype and the association of VLSFAs and the lower-risk phenotype warrant further research to explore potentially relevant new risk pathways in HFpEF.

## 7. Conclusions

In HFpEF patients, higher blood levels of C14:0 and C16:0 were associated with cardiometabolic risk factors. Contrarily, C18:0 and the three VLSFA were associated with a lower cardiometabolic risk profile. FA-based biomarkers warrant further investigation as prognostic markers in HFpEF.

## Figures and Tables

**Figure 1 biomedicines-10-02296-f001:**
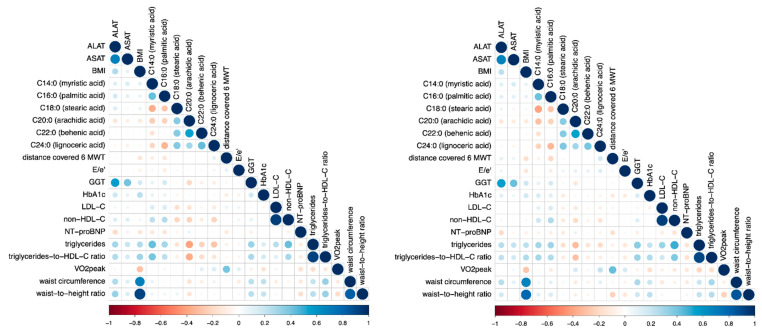
Correlation plot showing correlations between individual SFA and patient characteristics at baseline (**left**) and after 12-month follow-up (**right**). Blue color indicates a positive association; orange color indicates an inverse association. Higher color intensity and larger circles indicate a stronger association and lower color intensity, while smaller circles indicate a weaker association. Abbreviations: C14:0 (myristic acid), C16:0 (palmitic acid), C18:0 (stearic acid), C20:0 (arachidic acid), C22:0 (behenic acid), C24:0 (lignoceric acid), NT-proBNP (N-terminal pro–braintype natriuretic peptide), GGT (γ-glutamyltransferase), ASAT (aspartate aminotransaminase), ALAT (alanine aminotransaminase), E/e´ (diastolic function), VO2peak (maximum exercise capacity).

**Figure 2 biomedicines-10-02296-f002:**
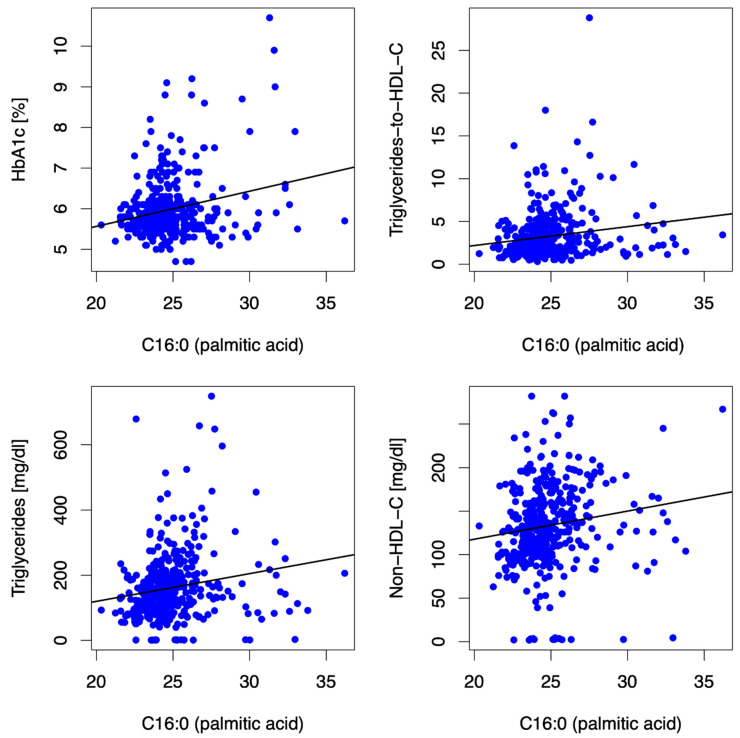
Scatter plots showing correlations between the LCSFA palmitic acid (C16:0) and HbA1c, triglycerides-to-HDL-C ratio, triglycerides, and non-HDL-C at baseline. The SFA palmitic acid (C16:0) was directly associated with HbA1c, triglycerides-to-HDL-C ratio, triglycerides, and non-HDL-C at baseline.

**Figure 3 biomedicines-10-02296-f003:**
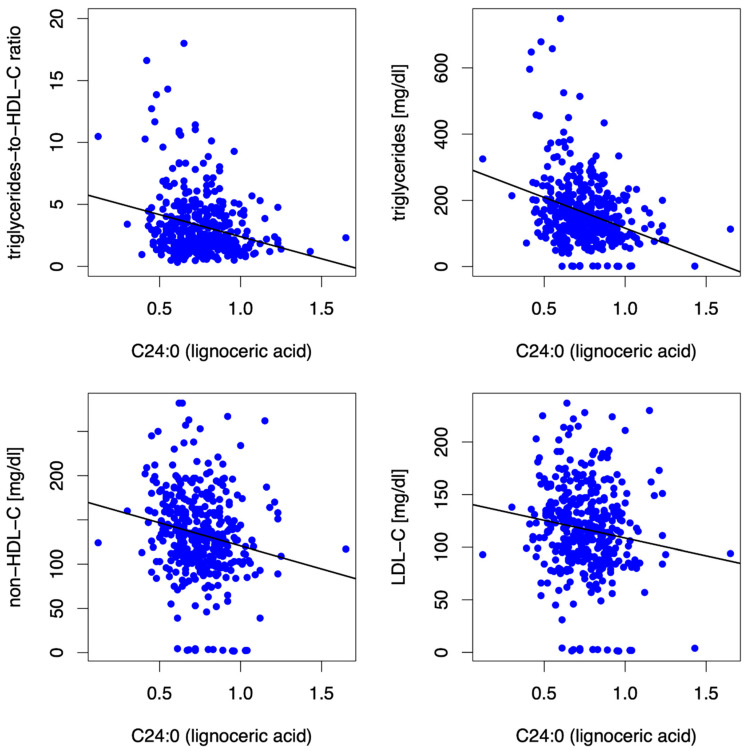
Scatter plots showing correlations between VLSFA lignoceric acid (C24:0) and triglycerides-to-HDL-C ratio, triglycerides, non-HDL-C, and LDL-C at baseline. The SFA lignoceric acid (C24:0) was inversely associated with triglycerides-to-HDL-C ratio, triglycerides, non-HDL-C, and LDL-C at baseline.

**Figure 4 biomedicines-10-02296-f004:**
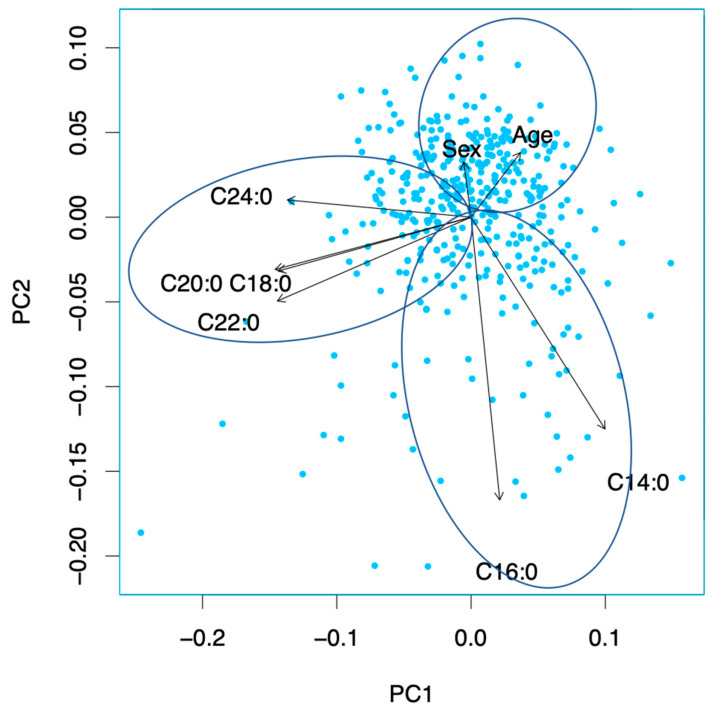
Principal component analysis. The first and second PCs are shown as biplot. The sum of variance of the first two principal components (PCs) was relatively low at 49.2%. The PCA shows that there are two variable groups, C14:0 and C16:0, as well as C18:0, C20:0, C22:0, and C24:0, which have a similar information content within the group. This is the same for age and sex. Abbreviations: C14:0 (myristic acid), C16:0 (palmitic acid), C18:0 (stearic acid), C20:0 (arachidic acid), C22:0 (behenic acid), C24:0 (lignoceric acid).

**Table 1 biomedicines-10-02296-t001:** Baseline characteristics.

Characteristics ^a^	Total (*n* = 404)
**Demographics**	
Age, mean (SD), y	67 (8)
Female	212 (53)
**Laboratory measures**	
HbA1c (%)	6.0 (0.8)
LDL-C (mg/dl)	117 (42)
HDL-C (mg/dl)	56 (18)
Triglycerides (mg/dl)	161 (103)
Non-HDL-C (mg/dl)	133 (47)
TG/HDL-C ratio	3.3 (2.8)
NT-proBNP, median (IQR), ng/L	158 (82–298)
C14:0 (myristic acid) (%)	0.69 (0.26)
C16:0 (palmitic acid) (%)	24.89 (2.17)
C18:0 (stearic acid) (%)	13.87 (1.35)
C20:0 (arachidic acid) (%)	0.18 (0.03)
C22:0 (behenic acid) (%)	0.5 (0.12)
C24:0 (lignoceric acid) (%)	0.75 (0.18)
**Medical history**	
Hospitalization for heart failure in past 12 months	149 (37)
Hypertension	370 (92)
Diabetes mellitus	66 (16)
Atrial fibrillation	65 (16)
**Physical examination, mean (SD)**	
Body mass index ^b^	28.9 (3.6)
Waist circumference, (cm)	98.1 (11.0)
In men	103.7 (9.0)
In women	93.1 (10.3)
Waist-to-height ratio	0.49 (0.1)
Systolic blood pressure, mm Hg	135 (18)
Diastolic blood pressure, mm Hg	79 (11)
Heart rate, /min	66 (11)
**Signs and symptoms**	
**NYHA functional class**	
II	350 (87)
III	54 (13)
Peripheral edema	160 (40)
Nocturia	325 (80)
Paroxysmal nocturnal dyspnea	66 (16)
Nocturnal cough	61 (15)
Fatigue	241 (60)
**Current medications**	
ACE inhibitors/angiotensin receptor antagonists	310 (77)
Betablockers	290 (72)
Diuretics	213 (53)
Calcium antagonists	97 (24)
Lipid-lowering drugs	221 (55)
**Echocardiography, mean (SD)**	
LV ejection fraction, %	68 (8)
LV diameter (end diastolic), mm	46.5 (6.2)
LV diameter (end systolic), mm	25.3 (6.4)
LV mass index, g/m^2^	114.15 (45.53)
Left atrial volume index, mL/m^2^	43.1 (41.6)
E-wave velocity, cm/s	73 (20)
Medial e’ wave velocity, cm/s	5.9 (1.3)
E/e’	7.1 (1.5)
E/A velocity ratio	0.91 (0.33)
Isovolumic relaxation time, ms	88 (26)
Deceleration time, ms	243 (63)
**Grade of diastolic dysfunction, No. (%)**	
I	295 (73)
II	81 (20)
III	4 (1)
IV	3 (1)

Abbreviations: ACE, angiotensin-converting enzyme; IQR, interquartile range; NT-proBNP, N-terminal pro–braintype natriuretic peptide; NYHA, New York Heart Association; A, peak atrial transmitral ventricular filling velocity; e’, early diastolic tissue Doppler velocity; E, peak early transmitral ventricular filling velocity. ^a^ Data are expressed as No. (%) unless otherwise specified. ^b^ Body mass index is defined as weight in kilograms divided by height in meters squared.

**Table 2 biomedicines-10-02296-t002:** Associations between individual SFA and patient characteristics at baseline.

		C14:0	C16:0	C18:0	C20:0	C22:0	C24:0
LDL-C	r ^§^	**0.145**	**0.148**	**−0.178**	**−0.153**	−0.059	**−0.12**
	*p*-value *	**0.005**	**0.004**	**0.001**	**0.003**	0.255	**0.021**
non-HDL-C	r ^§^	**0.234**	**0.257**	**−0.212**	**−0.23**	**−0.107**	**−0.227**
	*p*-value *	**<0.001**	**<0.001**	**<0.001**	**<0.001**	**0.039**	**<0.001**
triglycerides	r ^§^	**0.428**	**0.299**	**−0.155**	**−0.407**	**−0.232**	**−0.283**
	*p*-value *	**<0.001**	**<0.001**	**0.003**	**<0.001**	**<0.001**	**<0.001**
triglycerides-to-HDL-C ratio	r ^§^	**0.393**	**0.239**	−0.097	**−0.373**	**−0.19**	**−0.15**
	*p*-value *	**<0.001**	**<0.001**	0.061	**<0.001**	**<0.001**	**0.004**
HbA1c	r ^§^	0.089	0.083	−0.053	0.036	0.016	−0.035
	*p*-value *	0.087	0.112	0.306	0.489	0.76	0.501
ASAT	r ^§^	**0.137**	**0.107**	0.038	−0.092	−0.029	−0.084
	*p*-value *	**0.008**	**0.04**	0.467	0.078	0.576	0.105
ALAT	r ^§^	0.122	**0.166**	**0.129**	**−0.143**	−0.023	−0.047
	*p*-value *	0.019	**0.001**	**0.013**	**0.006**	0.656	0.371
GGT	r ^§^	0.095	**0.163**	−0.027	**−0.206**	−0.091	**−0.102**
	*p*-value *	0.069	**0.002**	0.6	**<0.001**	0.081	**0.049**
BMI	r ^§^	0.109	0.085	0.081	**−0.125**	−0.024	−0.041
	*p*-value *	0.036	0.101	0.119	**0.016**	0.651	0.435
waist circumference	r ^§^	0.08	0.036	0.024	**−0.159**	−0.098	0.078
	*p*-value *	0.124	0.493	0.645	**0.002**	0.059	0.135
waist-to-height ratio	r ^§^	0.068	0.06	0.076	**−0.151**	−0.044	0.038
	*p*-value *	0.191	0.246	0.146	**0.004**	0.394	0.462
distance covered 6MWT	r ^§^	−0.099	**−0.14**	0.008	0.011	−0.009	**0.105**
	*p*-value *	0.058	**0.007**	0.885	0.83	0.864	**0.044**
VO2peak	r ^§^	−0.032	0.042	0.027	−0.002	**0.112**	**0.14**
	*p*-value *	0.538	0.422	0.608	0.968	**0.031**	**0.007**
E/e‘	r ^§^	−0.045	−0.109	0.036	0.026	0.035	0.052
	*p*-value *	0.384	0.037	0.488	0.617	0.504	0.321
NT-proBNP	r ^§^	**−0.138**	**−0.147**	−0.058	0.075	−0.009	−0.027
	*p*-value *	**0.008**	**0.005**	0.262	0.151	0.867	0.607

Abbreviations: C14:0 (myristic acid), C16:0 (palmitic acid), C18:0 (stearic acid), C20:0 (arachidic acid), C22:0 (behenic acid), C24:0 (lignoceric acid), NT-proBNP (N-terminal pro–braintype natri-uretic peptide), GGT (γ-glutamyltransferase), ASAT (aspartate aminotransaminase), ALAT (ala-nine aminotransaminase), E/e´ (diastolic function), VO2peak (maximum exercise capacity). Significant values are in bold. * All tests were performed 2-sided. r^§^ refers to Spearman’s correlation coefficient.

**Table 3 biomedicines-10-02296-t003:** Associations between individual SFA and patient characteristics at 12 mFU.

		C14:0	C16:0	C18:0	C20:0	C22:0	C24:0
LDL-C	r ^§^	0.102	**0.151**	−0.085	**−0.126**	−0.044	−0.085
	*p*-value *	0.059	**0.005**	0.116	**0.02**	0.421	0.118
non-HDL-C	r ^§^	**0.185**	**0.265**	**−0.122**	**−0.196**	−0.073	**−0.154**
	*p*-value *	**0.001**	**<0.001**	**0.024**	**<0.001**	0.181	**0.004**
triglycerides	r ^§^	**0.291**	**0.306**	**−0.172**	**−0.32**	**−0.153**	**−0.196**
	*p*-value *	**<0.001**	**<0.001**	**0.001**	**<0.001**	**0.005**	**<0.001**
triglycerides-to-HDL-C ratio	r ^§^	**0.285**	**0.269**	**−0.115**	**−0.292**	**−0.109**	−0.106
	*p*-value *	**<0.001**	**<0.001**	**0.033**	**<0.001**	**0.045**	0.051
HbA1c	r ^§^	**0.146**	0.116	**−0.114**	−0.048	−0.048	−0.062
	*p*-value*	**0.007**	0.032	**0.036**	0.373	0.373	0.255
ASAT	r ^§^	**0.148**	0.102	−0.046	**−0.119**	0.024	−0.061
	*p*-value *	**0.006**	0.061	0.393	**0.028**	0.653	0.263
ALAT	r ^§^	0.11	**0.112**	0.074	**−0.139**	0.026	0.039
	*p*-value *	0.042	**0.038**	0.174	**0.01**	0.636	0.475
GGT	r ^§^	0.095	**0.161**	0.033	**−0.133**	−0.025	−0.014
	*p*-value *	0.078	**0.003**	0.544	**0.014**	0.651	0.803
BMI	r ^§^	0.095	0.094	0.041	**−0.126**	−0.039	−0.028
	*p*-value *	0.081	0.084	0.451	**0.02**	0.469	0.603
waist circumference	r ^§^	0.061	0.1	0.046	**−0.114**	−0.07	0.103
	*p*-value *	0.262	0.064	0.399	**0.036**	0.199	0.058
waist-to-height ratio	r ^§^	**0.117**	**0.149**	0.032	−0.093	−0.053	−0.009
	*p*-value *	**0.031**	**0.006**	0.557	0.086	0.327	0.866
distance covered 6MWT	r ^§^	−0.076	**−0.209**	0.036	0.022	0.018	**0.176**
	*p*-value *	0.159	**<0.001**	0.508	0.687	0.738	**0.001**
VO2peak	r ^§^	−0.037	−0.111	−0.018	**0.157**	0.046	**0.214**
	*p*-value *	0.494	0.041	0.735	**0.004**	0.396	**<0.001**
E/e‘	r ^§^	−0.028	−0.101	0.064	0.093	0.087	**0.123**
	*p*-value *	0.608	0.062	0.24	0.087	0.111	**0.023**
NT-proBNP	r ^§^	−0.071	−0.098	**−0.111**	0.077	−0.021	−0.067
	*p*-value *	0.192	0.072	**0.041**	0.155	0.701	0.219

Abbreviations: C14:0 (myristic acid), C16:0 (palmitic acid), C18:0 (stearic acid), C20:0 (arachidic acid), C22:0 (behenic acid), C24:0 (lignoceric acid), NT-proBNP (N-terminal pro–braintype na-tri-uretic peptide), GGT (γ-glutamyltransferase), ASAT (aspartate aminotransaminase), ALAT (ala-nine aminotransaminase), E/e´ (diastolic function), VO2peak (maximum exercise capacity). Significant values are in bold. * All tests were performed 2-sided. r^§^ refers to Spearman’s correlation coefficient.

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
