# Peer review of "Saturated Fatty Acid Blood Levels and Cardiometabolic Phenotype in Patients with HFpEF: A Secondary Analysis of the Aldo-DHF Trial"

_biomedicines, 2022, doi:10.3390/biomedicines10092296_

Round 1

Reviewer 1 Report

The manuscript by Lechner K et al addresses the differential levels of saturated fatty acids (SFAs: C14:0, C16:0, C18.0, C20:0, C22:0 and C24:0), according to chromatographic analysis, as molecular targets linked to heart failure with preserved ejection fraction (HFpEF). Increasing evidence suggests that there is differential profile based in SFAs and clinic characteristics; highlighting its association with the risk cardiometabolic phenotype. Here, the authors investigate the regulation of these SFAs in HFpEF-patients at baseline and after 12-months-follow-up (12mFU).

Major concerns

1.- The manuscript presented by Lechner K et al  is based on clinic characteristics in HFpEF –patients at baseline However, the authors did not show the clinical characteristic of HFpEF –patients (12mFU) or a comparative study with patients with heart failure with reduced ejection fraction (HFrEF) to validate the SFA correlations, as important clinical endpoint in their findings. The authors evaluated the effect of 12 months with aldosterone receptor blockade treatment on diastolic function (E/e’) and maximal exercise capacity (VO2peak) in HFpEF –patients. Which was the relevance by the authors to study waist circumference, waist-to height ratio and distance covered 6 MWT?

2.- The authors displayed the correlations among SFAs and clinical characteristics in Figure 1 and Figure 2. To facilitate the comparison between HFpEF-patients at baseline and 12mFU, the authors should show these results in the same Figure. The correlations should be ordered in same pattern in both groups in order to appreciate the differential spot plots. On the other hand, in Figure 1 and Figure 2, the correlations among SFAs are similar. The authors should show a comparative analysis of SFA concentrations at baseline and 12mFU.

3.- The authors aim to displaying  the correlations among SFAs and clinical characteristics in Table 2 and 3, however  they are very low and its interpretation could be overestimated. The authors should analyze the differential clinical profile based on median of SFAs as cut-off of to HFpEF at baseline and 12 mFU.

4.- According to the manuscript, the sex covariate was not explained in detail. The authors should analyse their findings in men versus women population, providing strength to the manuscript.   

5. - To better understand the translational relevance of the present manuscript, the authors should more extensively discuss their findings in the context of previous studies focused in C14:0 and C16:0 related to high risk of cardiovascular disease (Jordi Mayneris-Perxachs et al, Clin Nutr. 2014 Feb;33(1):90-7) and C16:0, C22:0, C24:0 associated to risk of specific mortality (Maura E Walker et al, J Nutr. 2020 Nov 19;150(11):2994-3004).

Minor concerns

Methods

1.- The experiment results of SFAs should not be shown in Table 1. he population study should be described in detail. Which were the criteria of inclusion and exclusion?

2.- The description of laboratory methods: ALDO-DHF trial should be included in HS-Omega-3 Index methodology.

3.- The associations of individual SFAs (Table 2 and 3) should be mentioned in results section.

Results

1.-  The authors did not demonstrate the effect of dairy fat intake and could have a bias interpretation in their results.  Is there a standard diet or protocol to control this bias?

2.- The authors should better explain the Figure 5 and its useful to discussion and conclusion sections

Reviewer 2 Report

This work provided associations of individual whole blood SFA proportions with cardiometabolic risk factors, exercise capacity, echocardiographic markers of left ventricular diastolic function, and neurohumoral activation in patients with HFpEF. The most consistent inverse association of whole blood VLSFAs was observed with lipid risk markers, in particular triglycerides. No consistent significant associations and/or specific clusters were found between cardiac function and/or neurohumoral activation and blood SFA concentrations in this analysis, consistent with our previous analysis investigating the associations of individual n-3 FA and echocardiographic markers of left ventricular diastolic function. Individual blood SFAs could be broadly categorized into two main SFA conglomerations which are differentially linked to biomarkers and anthropometric markers indicative of higher-/lower risk cardiometabolic phenotype in HFpEF patients. Finally, the author concluded higher blood levels of C14:0 and C16:0 were associated with cardiometabolic risk factors. Contrarily, C18:0 and the three VLSFA were associated with a lower cardiometabolic risk profile in the patients with HFpEF. This work provided important information for patients with HFpEF.

One minor comment

The authors may eliminate the red lines and yellow colors in Table 2, and Table 3.

Round 2

Reviewer 1 Report

In this new version, the authors have addressed most of the concerns raised by the reviewer, expanded the results and modified the discussion according to the stated queries.